# Year-round temporal stability of a tropical, urban plant-pollinator network

**Alyssa B. Stewart** [1]*, **Pattharawadee Waitayachart**[2]

**1** Department of Plant Science, Faculty of Science, Mahidol University, Bangkok, Thailand, **2** Department of Biology, Faculty of Science, Mahidol University, Bangkok, Thailand

* alyssa.ste@mahidol.edu

**Data Availability Statement:** Data was uploaded to a public repository (Mendeley Data): https://dx.doi.org/10.17632/4vbdh6j2tx.1.

**Funding:** This research was supported by the Thailand Research Fund (grant MRG6080124 to ABS) and Mahidol University (Mentorship Grant

## Abstract

Plant-pollinator interactions are known to vary across time, both in terms of species composition and the associations between partner species. However, less is known about tropical pollination networks, and tropical urban parks provide a unique opportunity to study network stability in an environment where temperature and floral resources are relatively constant due to both the tropical climate as well as park horticulture. The objectives of this study were thus to examine the interactions between flowering plants and their potential pollinators in a large, tropical city (Bangkok, Thailand) across 12 consecutive months, and to assess the stability of network properties over time. We conducted monthly pollinator observations at 9 parks spaced throughout the city, and collected data on temperature, precipitation, floral abundance and floral species richness. We found that neither pollinator abundance nor richness varied significantly across months when all parks were pooled. However, pollinator abundance was significantly influenced by floral abundance, floral richness, and their interaction, and pollinator richness was significantly influenced by floral richness and precipitation. Finally, we found that network properties did not change across months, even as species composition did. We conclude that the year-round constancy of floral resources and climate conditions appear to create a network in dynamic equilibrium, where plant and pollinator species compositions change, but network properties remain stable. The results of this study provide useful information about how tropical pollinators respond to urban environments, which is particularly relevant given that most urban development is predicted to occur in the tropics.

## Introduction

While the importance of plant-pollinator interactions has long been recognized [1,2], we have only recently developed the computational methods needed to analyze entire networks, and over extended periods rather than at a single point in time [3,4]. Analyzing complete pollination networks provides a more accurate understanding of the community, since both plants and pollinators typically interact with more than one partner [5–7]. Moreover, studying changes in pollination networks over time can (1) improve our understanding of pollination services, as most angiosperm species rely on animal pollinators [8,9]; (2) provide clarity on

co-awarded to ABS). The funders had no role in study design, data collection and analysis, decision to publish, or preparation of the manuscript.

**Competing interests:** The authors have declared that no competing interests exist.

whether species are specialists, generalists, or sequential specialists [10,11]; (3) facilitate predictions of how plants, pollinators, and their interactions will respond to climate change [12]; and (4) provide critical information that can help with conservation efforts of both pollinators, which are undergoing population declines worldwide, and the plants that depend on them for reproduction [13,14].

Previous studies examining pollination networks over time have found a range of temporal patterns, from networks that appear highly stable to those that are highly dynamic [4]. Yet most work to date has been conducted in temperate or arctic regions [10,15–24] and we still know very little about the stability of tropical pollination networks (but see [25,26]). Thus, while ecologists are interested in finding universal patterns and mechanisms behind plant-animal networks [3], we still lack data for tropical systems.

Additionally, given the rate at which humans are changing the environment, it is important to study interactions not only in natural habitats, but also in human-modified landscapes [27]. For example, global urban land area is predicted to triple in size by 2030, compared to 2000, and most urban expansion is projected to occur in tropical areas [28]. Therefore, studying plant-pollinator interactions in cities can provide valuable information about how pollinators fare in one of the few habitat types that is expected to increase in size in the near future [27,28]. Finally, the abundance of floral resources found year-round in tropical cities [29], often due to the mix of native and exotic plant species found in cultivated parks, provides a unique opportunity to study plant-pollinator interactions in a relatively stable environment. Therefore, the objectives of this study were to examine the interactions between flowering plants and their potential pollinators at public parks in Bangkok, Thailand across 12 consecutive months, and to assess the stability of network properties throughout this period.

## Methods

### Study area

Data were collected in Bangkok, Thailand between December 2017—November 2018. Bangkok is the most populated city in Thailand (over 9.6 million residents according to the 2010 census) with very little natural habitat; nearly all vegetation is cultivated and managed [30]. The climate is tropical; average monthly temperatures range between 22–32˚C and average annual rainfall in the area ranges from 1,100–1,600 mm [31]. We collected data from 9 parks spaced throughout the city (S1 Fig). These nine parks were selected because they are all at least 1 km apart, and represent a diversity of park sizes (S1 Fig) and management intensities [30]. All nine parks were regularly watered, but experienced different degrees of floral rotation; some parks periodically brought in new flowering plant species as previously planted species dropped their flowers, while other parks did not rotate plant species at all (A. Stewart, pers. obs.). It is common for parks in Bangkok to have both native and exotic plant species [30].

### Data collection

Monthly pollinator observations were conducted over 12 consecutive months (December 2017—November 2018). The order in which parks were visited during each month was random, and sampling was only conducted on sunny days. All sampling at each park was conducted within a single day, and up to three parks were observed in a single day. During each month, the first and last parks were sampled within 16–25 days of each other. During each sampling event, we conducted 15-minute observations at all of the most abundant plant species that were in flower. In order to be efficient with sampling, we only observed plant species with at least 20 flowers or inflorescences; this resulted in approximately 80–90% of the flowering plant community being sampled at each park in each month. We chose this sampling

method because we wanted to maximize the number of flowering plant species sampled in order to track floral resource availability across months, and to record as many plant-pollinator interactions as possible, but were limited by manpower. For each plant species observed, a 2 x 2 m plot with abundant flowers of the focal species was selected, and all animals that contacted floral reproductive structures while visiting flowers (i.e., potential pollinators) were recorded. As we did not quantify actual pollen transfer, we cannot confirm that all recorded taxa were true pollinators; it is likely that observed animal taxa differ in pollination effectiveness, and some taxa may not have contributed to pollination at all. We chose to conduct non-destructive pollinator sampling (i.e., relying on visual observation) because collecting individuals may have reduced pollinator abundance and richness at adjacent plots within the park. Unknown pollinators were either photographed or collected with a net, and insects were identified with the help of local field guidebooks and entomologists (see acknowledgments). Plots were not fixed; they varied by month according to which plant species were flowering (mean ± SE plant species sampled per park per month: 13.8 ± 0.7 species; range: 1–33 species). Permission to work with insects was granted by MUSC-IACUC (Faculty of Science, Mahidol University–Institute Animal Care and Use Committee; license number MUSC60-038-388).

We also obtained data on average monthly temperature (hereafter, "temperature"), total monthly precipitation (hereafter, "precipitation"), floral abundance, and floral species richness. Temperature and precipitation data were acquired from the Thai Meteorological Department (www.tmd.go.th). Floral abundance was determined at each plot as the number of flowers per 2 x 2 m plot; these values were then averaged to obtain a mean number of flowers per plot for each park in each month. Floral richness was determined as the number of flowering plant species sampled at each park in each month.

## Data analysis

All analyses were conducted in R 3.6.0 [32]. We used linear mixed modelling (LMM; package "lme4") to examine which predictors influenced pollinator abundance and richness. The response variables examined were total pollinator abundance, total pollinator richness, and the abundance and richness of the most common insect orders (Hymenoptera, Lepidoptera, and Diptera). The fixed factors tested were month, temperature, precipitation, temperature x precipitation, floral abundance, floral richness, and floral abundance x floral richness; park was included as a random factor. QQ plots (package "stats") were used to check the normality of the residuals. In the model prediction graphs, when two explanatory predictors were found to be significant, one predictor was plotted along the x-axis and the second predictor was represented using various colors; this method also clearly shows when the two predictors have a significant interaction (non-parallel lines) or not (parallel lines).

We also examined pollination networks using the package "bipartite" [33,34]. One overall network was created using data from all parks and all months, and we also created separate networks for each month at each park to examine changes in the pollinator community over time (S2 Fig). We examined network properties at the network level (connectance, weighted connectance, links per species, number of compartments, and Shannon's diversity), group level (number of species within each trophic level, mean number of links within each trophic level, and niche overlap within each trophic level), and species level (normalized degree averaged across all species, and paired differences index averaged across all species within each trophic level). Descriptions of each network property are provided in S1 Text. We examined whether each network property changed over time using LMM, where month was a fixed factor and park was a random factor. We also calculated species turnover (proportion of species lost or gained between two consecutive months) for both plants and pollinators using the "codyn" package [35].

## Results

### Pollinator abundance and richness

Over all 12 months of data collection (Fig 1), we observed 8,053 potential pollinators comprising 58 taxa from 4 orders (S1 Table) visiting 136 plant taxa from 48 orders (S2 Table). Hymenoptera were by far the most abundant (95.2%), followed by Diptera (2.5%), Lepidoptera (1.8%), and Hemiptera (0.5%). We observed 23 taxa of Hymenoptera, 23 species of Lepidoptera, 11 taxa of Diptera, and 1 taxon of Hemiptera. The most abundant taxa were *Tetragonula* stingless bees (60.2%), *Apis florea* (Fabricius, 1787) (17.6%), *A. cerana* (Fabricius, 1793) (6.9%), *A. dorsata* (Fabricius, 1793) (4.9%), *Lasioglossum* sweat bees (3.7%), and Tephritidae flies (2.1%); all other taxa comprised less than one percent of total abundance (S1 Table).

When examining total pollinator abundance, we found that neither month, temperature, precipitation, nor temperature x precipitation had a significant effect, but both floral abundance and floral richness had positive effects, and their interaction was significant as well (Table 1; Fig 2A). For total pollinator richness, both precipitation and floral richness had significant positive effects (Table 1; Fig 2B). Hymenoptera abundance was significantly influenced by floral abundance (positive effect), floral richness (positive effect), and their interaction (Table 1; Fig 2C), while Hymenoptera richness was influenced by precipitation and floral richness (positive effects; Table 1; Fig 2D). Both Lepidoptera abundance and richness were positively influenced by floral richness (Table 1; Fig 2E and 2F). For Diptera abundance and richness, none of the predictors included in the model were significant (Table 1).

### Pollination networks

The overall plant-pollinator network (using data from all parks across all months) had a connectance of 0.057, a weighted connectance of 0.092, an average of 2.05 links per species, 3 compartments, and a Shannon diversity index of 4.46. At the group level, pollinators had an average of 5.29 links, while plants had an average of 5.96 links. The average number of shared partners was 0.54 for pollinators and 1.02 for plants. Niche overlap was 0.081 for pollinators and 0.44 for plants. When examining whether network properties changed over time, we found that none of the tested properties varied by month (Table 2; S2 Fig). We did, however, find that both plant and pollinator composition changed across months; species turnover between consecutive months was close to 50% for both plants (mean ± SE: 0.45 ± 0.04) and pollinators (0.46 ± 0.04).

## Discussion

Neither total pollinator abundance nor richness varied significantly by month. The steady numbers of both individuals and species likely reflect the fact that tropical areas have mild climates, which allows pollinators to be active year-round [8,36]. Yet even tropical environments can have seasonal fluctuations in insect abundance and diversity, because floral (food) availability is often markedly different between the rainy and dry seasons in most natural landscapes [26]. Therefore, the second factor contributing to constant pollinator abundance and richness is likely the constancy of floral resources found in our study parks. Urban parks often favor plant species with showy flowers, and cultivate a mix of native and exotic species that provide abundant floral resources year-round [29,37,38]. Indeed, of the 136 plant taxa observed in this study, 73 species were exotic and 47 were native; the remaining 16 could not be verified (S2 Table). The benefits of reliable floral resources in urban habitats have also been demonstrated in temperate bees [23] and tropical butterflies [39], where the abundance and richness of these taxa were less variable in urban environments than natural habitats.

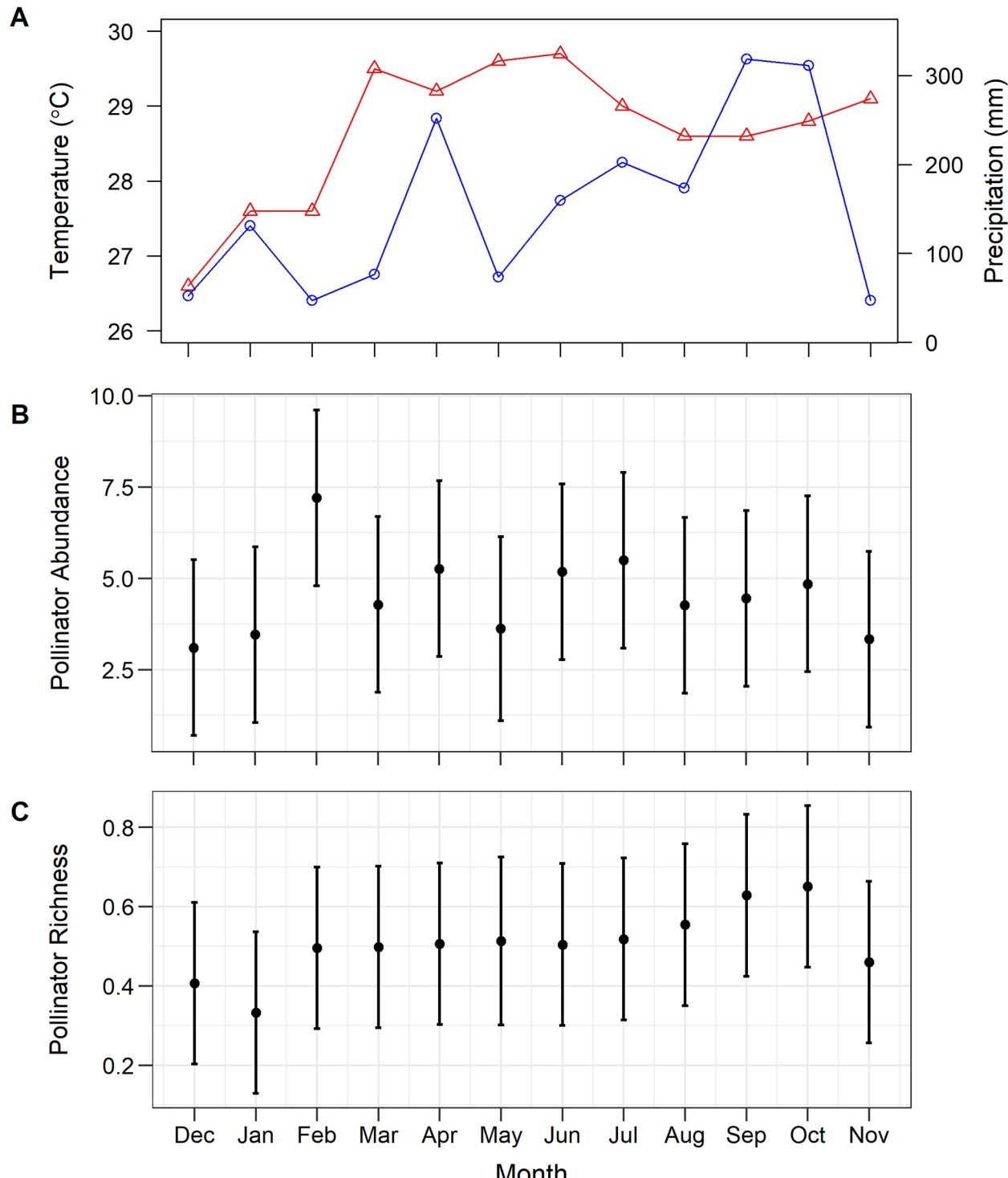

**Fig 1. Temporal variation in weather and pollinator communities in Bangkok, Thailand over 12 months (December 2017—November 2018).**
(A) Average monthly temperature (triangles, red line) and total monthly precipitation (circles, blue line). Neither (B) pollinator abundance nor (C) pollinator species richness varied significantly across months; points and error bars represent mean ± SE.

**Table 1. Results of linear mixed modelling examining the effect of 7 predictors on pollinator abundance and species richness.**

| | All pollinators | | Hymenoptera | | Lepidoptera | | Diptera | |
|---|---|---|---|---|---|---|---|---|
| | **Abund.** $R_m^2 = 0.19$ $R_c^2 = 0.40$ | **Rich.** $R_m^2 = 0.20$ $R_c^2 = 0.52$ | **Abund.** $R_m^2 = 0.19$ $R_c^2 = 0.04$ | **Rich.** $R_m^2 = 0.21$ $R_c^2 = 0.42$ | **Abund.** $R_m^2 = 0.10$ $R_c^2 = 0.34$ | **Rich.** $R_m^2 = 0.08$ $R_c^2 = 0.36$ | **Abund.** n/a | **Rich.** n/a |
| Month | $\chi_1^2 = 0.09$ P = 0.76 | $\chi_1^2 = 0.16$ P = 0.69 | $\chi_1^2 = 0.09$ P = 0.77 | $\chi_1^2 = 0.43$ P = 0.51 | $\chi_1^2 = 2.13$ P = 0.14 | $\chi_1^2 = 2.53$ P = 0.11 | $\chi_1^2 = 3.43$ P = 0.06 | $\chi_1^2 = 3.04$ P = 0.08 |
| Temperature | $\chi_1^2 = 0.22$ P = 0.64 | $\chi_1^2 = 0.59$ P = 0.44 | $\chi_1^2 = 0.24$ P = 0.62 | $\chi_1^2 = 1.33$ P = 0.25 | $\chi_1^2 = 0.61$ P = 0.43 | $\chi_1^2 = 1.01$ P = 0.31 | $\chi_1^2 = 0.03$ P = 0.87 | $\chi_1^2 = 0.09$ P = 0.77 |
| Precipitation | $\chi_1^2 = 2.75$ P = 0.10 | $\chi_1^2 = 4.23$ **P = 0.04** | $\chi_1^2 = 2.70$ P = 0.10 | $\chi_1^2 = 4.53$ **P = 0.03** | $\chi_1^2 = 0.24$ P = 0.62 | $\chi_1^2 = 0.003$ P = 0.96 | $\chi_1^2 = 0.07$ P = 0.80 | $\chi_1^2 = 0.04$ P = 0.84 |
| Temperature x precipitation | $\chi_1^2 = 1.29$ P = 0.26 | $\chi_1^2 = 0.01$ P = 0.92 | $\chi_1^2 = 1.52$ P = 0.22 | $\chi_1^2 = 0.20$ P = 0.66 | $\chi_1^2 = 1.50$ P = 0.22 | $\chi_1^2 = 1.47$ P = 0.23 | $\chi_1^2 = 1.29$ P = 0.26 | $\chi_1^2 = 0.09$ P = 0.77 |
| Floral abundance | $\chi_1^2 = 6.72$ **P = 0.03** | $\chi_1^2 = 1.07$ P = 0.30 | $\chi_1^2 = 6.77$ **P = 0.03** | $\chi_1^2 = 1.59$ P = 0.21 | $\chi_1^2 = 0.01$ P = 0.92 | $\chi_1^2 = 0.23$ P = 0.63 | $\chi_1^2 = 1.36$ P = 0.24 | $\chi_1^2 = 0.84$ P = 0.36 |
| Floral richness | $\chi_1^2 = 13.7$ **P = 0.001** | $\chi_1^2 = 8.63$ **P = 0.003** | $\chi_1^2 = 13.6$ **P = 0.001** | $\chi_1^2 = 8.90$ **P = 0.003** | $\chi_1^2 = 6.57$ **P = 0.01** | $\chi_1^2 = 5.28$ **P = 0.02** | $\chi_1^2 = 0.21$ P = 0.65 | $\chi_1^2 = 0.01$ P = 0.92 |
| Floral abundance x richness | $\chi_1^2 = 4.31$ **P = 0.04** | $\chi_1^2 = 0.38$ P = 0.54 | $\chi_1^2 = 4.39$ **P = 0.04** | $\chi_1^2 = 0.82$ P = 0.36 | $\chi_1^2 = 1.13$ P = 0.29 | $\chi_1^2 = 0.47$ P = 0.49 | $\chi_1^2 = 0.17$ P = 0.68 | $\chi_1^2 < 0.001$ P = 0.98 |

Separate analyses were conducted for total pollinator abundance, total pollinator richness, and the abundance and richness of each of the three most common insect orders observed (Hymenoptera, Lepidoptera, and Diptera). Significant predictors are highlighted in yellow with p-values in bold. Marginal ($R_m^2$) and conditional ($R_c^2$) $R^2$ values are listed for each final model.

The factors that significantly influenced total pollinator abundance were floral abundance, floral richness, and their interaction. Therefore, pollinator abundance was greatest when both floral abundance and richness were high, and pollinator abundance remained low (even at high floral abundances) when floral richness was low. It is important to note that our measures of pollinator abundance were obtained from our study plots of flowering plant species, and that many areas of the parks had substantially lower floral abundance; such areas would undoubtedly have fewer pollinators than was observed at our study plots. The importance of floral abundance and richness to pollinators seems to be relatively consistent worldwide and across diverse insect taxa [30,40–45]. Such findings are hardly surprising, given the importance of food resources in supporting pollinator communities [29,30,41–46].

However, our study did not find a significant effect of temperature or precipitation on total pollinator abundance, which is contradictory to several prior studies. For example, Andrade-Silva et al. [47] found that temperature was the most important determinant of euglossine bee abundance in a Brazilian forest, and Silva et al. [48] found that most orders of insects were influenced by temperature in a Brazilian savannah. However, Silva et al. [48] also explained that a likely mechanism behind their results was the increase of leaves and flowers during the transition from the dry to wet season. This justification is also consistent with our results; if insect abundance is primarily driven by food abundance, it seems logical that the year-round supply of flowers in our study parks are able to maintain high pollinator abundance across seasons. We also hypothesize that temperature and water availability in our study area were not variable enough to affect local pollinator abundance. The average monthly temperature during our study period ranged a mere 3 degrees (26.6–29.7°C), and plants within our study parks were never water limited due to regular watering by gardeners.

We also found that total pollinator richness was significantly influenced by floral richness and precipitation. Previous studies in both temperate [49] and tropical [25,50] regions have also found positive correlations between floral diversity and pollinator diversity. It is likely that

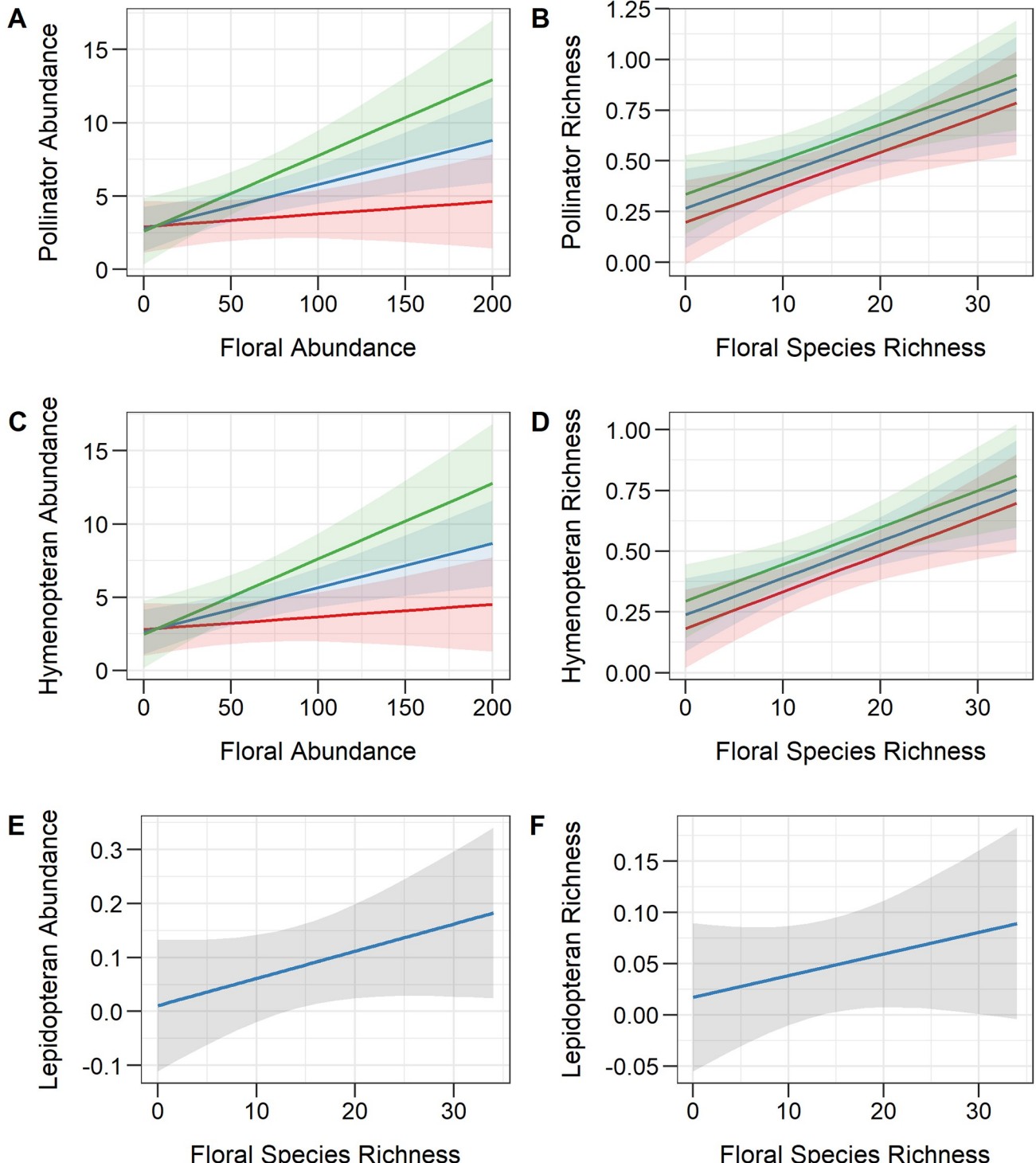

**Fig 2. Model predictions of pollinator abundance and richness in Bangkok, Thailand.** (A) Total pollinator abundance was significantly influenced by floral abundance (x-axis), floral richness (denoted by color; red: 6.6 spp., blue: 13.9 spp., green: 21.1 spp.), and their interaction. (B) Total pollinator richness was significantly influenced by floral richness (x-axis) and precipitation (denoted by color; red: 58 mm, blue: 155 mm, green: 251 mm). (C) Hymenopteran abundance was significantly influenced by floral abundance (x-axis), floral richness (denoted by color; red: 6.6 spp., blue: 13.9 spp., green: 21.1 spp.), and their interaction. (D) Hymenopteran richness was significantly influenced by floral richness (x-axis) and precipitation (denoted by color; red: 58 mm, blue: 155 mm, green: 251 mm). (E) Lepidopteran abundance and (F) Lepidopteran richness were significantly influenced by floral richness (x-axis).

**Table 2. Results of linear mixed modelling examining whether plant-pollinator network properties varied across 12 months (December 2017—November 2018).**

|  | Network property | Mean | SE | Chi-square | P |
|---|---|---|---|---|---|
| Network level | Connectance | 0.356 | 0.016 | $\chi_1^2 = 2.29$ | 0.13 |
|  | Weighted connectance | 0.209 | 0.008 | $\chi_1^2 = 2.10$ | 0.15 |
|  | Links per species | 0.845 | 0.018 | $\chi_1^2 = 1.85$ | 0.17 |
|  | Number of compartments | 1.987 | 0.106 | $\chi_1^2 = 1.11$ | 0.29 |
|  | Shannon's diversity | 1.796 | 0.063 | $\chi_1^2 = 0.64$ | 0.43 |
| Group level | Number of pollinator species | 5.228 | 0.304 | $\chi_1^2 = 0.40$ | 0.53 |
|  | Number of plant species | 6.582 | 0.341 | $\chi_1^2 = 0.17$ | 0.68 |
|  | Links per pollinator species | 3.614 | 0.224 | $\chi_1^2 = 0.39$ | 0.53 |
|  | Links per plant species | 1.927 | 0.095 | $\chi_1^2 = 2.07$ | 0.15 |
|  | Niche overlap among pollinators | 0.256 | 0.026 | $\chi_1^2 = 0.13$ | 0.72 |
|  | Niche overlap among plants | 0.433 | 0.029 | $\chi_1^2 = 0.52$ | 0.47 |
| Species level | Normalized degree | 0.356 | 0.016 | $\chi_1^2 = 2.29$ | 0.13 |
|  | Paired differences index (pollinators) | 0.923 | 0.007 | $\chi_1^2 = 2.33$ | 0.13 |
|  | Paired differences inex (plants) | 0.938 | 0.009 | $\chi_1^2 = 0.70$ | 0.40 |

Network property descriptions are provided in S1 Text. Mean and SE were calculated from 108 plant-pollinator networks (nine parks over 12 months).

the higher diversity of flowering plant species attracts a higher diversity of insect pollinators. Less conclusive is the effect of precipitation on pollinator richness. In this study, the rainiest months were September and October, which correspond with the greatest numbers of pollinator species observed. Precipitation likely had little effect on park vegetation and floral resources, given regular watering and park horticulture, but it possibly influenced other aspects of insect life history and/or behavior. Findings from previous studies are mixed. For example, studies of butterflies in Brazil have found the highest species richness to occur during the rainy season [51], during the dry season [25], and during the transition from rainy to dry season [52]. Therefore, it appears that the effects of precipitation on species richness are complex and likely linked to other factors as well, such as plant [51] and predator [52] composition.

When examining the three most common insect orders separately, we found that Hymenoptera results were consistent with the overall results, Lepidoptera abundance and richness were influenced only by floral richness, while Diptera abundance and richness were not explained by any of the tested predictors. Hymenoptera abundance was influenced by the same three factors as total pollinator abundance (floral abundance, floral richness, and their interaction); indeed, Hymenoptera abundance was the driving force behind our results for total pollinator abundance, since Hymenoptera accounted for 95% of all observed pollinators. Lepidoptera were found to forage on fewer plant species than Hymenoptera (S2 Fig), so it seems logical that their foraging is driven by high floral richness, as the presence of numerous floral species would increase the odds of there being at least one species attractive to butterflies. The lack of significant findings for Diptera may be due to small sample sizes, or it is possible that their abundance and richness are influenced by other factors not measured in this study. Previous studies have demonstrated that different species of Diptera [48] can peak during different seasons, resulting in unclear patterns at the level of order.

When examining plant-pollinator networks across 12 consecutive months, we found that network properties remained constant even as plant and pollinator composition changed. Our networks reflect the large majority of plant species found in the parks, although our inability to collect data on the rarest plant species may have influenced some network property estimates.

The temporal stability of pollination network properties has been highly investigated in recent years, across a broad range of taxa and landscapes; results vary from reporting temporally stable network properties [10,19–21] to highly dynamic network properties [15,16,18]. We suggest that the stability of network properties depends on both the timescale at which the network is analyzed, as well as the stability of environmental conditions. Networks examined over longer timescales (e.g., over multiple years) are more likely to reveal temporally stable network properties (as found by [10,19–21]), while networks examined over shorter timescales (e.g., within or between seasons) are more likely to find temporally dynamic network properties (as found by [15,16,18]). Moreover, the stability of network properties is likely correlated with the stability of environmental conditions. While our study spanned different seasons, it was conducted in a tropical, urban landscape where both temperature and floral resources remained relatively constant year-round. This reason may explain why we observed no differences in network properties across months, unlike previous year-long or season-long studies conducted in temperate [16] and arctic [15,18] environments.

## Conclusions

The findings from this study reveal that tropical urban parks are capable of supporting stable pollinator communities year-round. Even though our study was conducted in Bangkok, a city with little natural vegetation, the city's parks provide abundant floral resources throughout the year due to landscaping efforts, which favor plant species with showy flowers. However, it is important to note that (1) our networks included only the most abundant plant species, which may over- or underestimate some network properties, and (2) these parks are likely only suitable for pollinator species with generalist foraging and nesting habits, and that are tolerant of human activity. For example, we only observed 58 insect taxa visiting flowers, in contrast to a recent study in Thai mixed fruit orchards that recorded 316 insect pollinator taxa [53]. Moreover, our results suggest that the temporal stability of plant-pollinator network properties is driven by the stability of environmental conditions, including both climate and resource stability. In our study area, the constant floral resources and climate conditions throughout the year appear to create a network in dynamic equilibrium, where plant and pollinator species compositions change, but network properties remain stable year-round. These findings provide insight into how tropical pollinators respond to urban habitats, which will be useful as urban centers continue to grow world-wide.

## Supporting information

**S1 Fig. Map of study parks in Bangkok, Thailand.**
(PDF)

**S2 Fig. Plant-pollinator networks at each study park in Bangkok, Thailand over 12 months.**
(PDF)

**S1 Text. Descriptions of plant-pollinator network properties examined.**
(PDF)

**S1 Table. Pollinator species observed in Bangkok, Thailand between December 2017—November 2018.**
(PDF)

**S2 Table. Plant species at which pollinator observations were conducted (Bangkok, Thailand; December 2017—November 2018).**
(PDF)

## Acknowledgments

We thank Bangkok's Office of Public Parks for allowing us to conduct this research, and are grateful for all of the assistance provided by personnel at Benjakitti Park, Lumphini Park, Phaya Thai Pirom Garden, Santi Chai Prakan Public Park, Santiphap Park, Saranrom Park, Somdet Saranrat Maneerom Public Park, Vibhavadi Rangsit Forest Park, and Wachirabench-athat Park. We also thank Kaewagsorn Saowong and Narut Laonipon for helping with data collection, Natapot Warrit and Tom Stewart for helping identify insects, and Piriya Hassa for helping identify plants. Finally, we thank the two reviewers who provided thoughtful comments and suggestions for our manuscript.

## Author Contributions

**Conceptualization:** Alyssa B. Stewart.

**Data curation:** Alyssa B. Stewart, Pattharawadee Waitayachart.

**Formal analysis:** Alyssa B. Stewart.

**Funding acquisition:** Alyssa B. Stewart.

**Investigation:** Alyssa B. Stewart, Pattharawadee Waitayachart.

**Methodology:** Alyssa B. Stewart, Pattharawadee Waitayachart.

**Writing – original draft:** Alyssa B. Stewart.

**Writing – review & editing:** Alyssa B. Stewart, Pattharawadee Waitayachart.

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
