## [Decision Letter · Decision Letter 0]

6 Nov 2019

PONE-D-19-19692

Year-round temporal stability of a tropical, urban pollination network

PLOS ONE

Dear Dr. Stewart,

Thank you for submitting your manuscript to PLOS ONE. After careful consideration, we feel that it has merit but does not fully meet PLOS ONE’s publication criteria as it currently stands. Therefore, we invite you to submit a revised version of the manuscript that addresses the points raised during the review process.

This work has now been evaluated by two reviewers. Both reviewers find important merits in this study, however they also raised important concerns, and both of them think that the study only partially supports the conclusions and is technically sound. Reviewer I is concerned about the spatial variation of measurements, and raised questions about the type of parks, the study species and the sampling protocols. Reviewer II highlighted several limitations of the methodology, and have raised many concerns about the statistical analyses used. This reviewer also asks for further clarification of the effect of precipitation in parks that are regularly watered, and suggest the addition of references of pollinator abundance or species richness in natural habitats within the same region to evaluate the conservation value of these parks. Based on these evaluations, with which I concur, I would be happy to consider for further evaluation a substantially revised version of the manuscript that considers the suggestions and comments raised by the reviewers. If you decide to do this review, please include a response letter indicating your responses to the reviewer comments and the changes you have made in the manuscript.  If you disagree with a reviewer's point, explain why.

We would appreciate receiving your revised manuscript by Dec 21 2019 11:59PM. To enhance the reproducibility of your results, we recommend that if applicable you deposit your laboratory protocols in protocols.io, where a protocol can be assigned its own identifier (DOI) such that it can be cited independently in the future. For instructions see: http://journals.plos.org/plosone/s/submission-guidelines#loc-laboratory-protocols

We look forward to receiving your revised manuscript.

Kind regards,

Amparo Lázaro, PhD

Academic Editor

PLOS ONE

Journal Requirements:

1. We note that you have stated that you will provide repository information for your data at acceptance. Should your manuscript be accepted for publication, we will hold it until you provide the relevant accession numbers or DOIs necessary to access your data. If you wish to make changes to your Data Availability statement, please describe these changes in your cover letter and we will update your Data Availability statement to reflect the information you provide.

Reviewers' comments:

Reviewer's Responses to Questions

**Comments to the Author**

1. Is the manuscript technically sound, and do the data support the conclusions?

Reviewer #1: Partly

Reviewer #2: Partly

2. Has the statistical analysis been performed appropriately and rigorously? 

Reviewer #1: Yes

Reviewer #2: No

3. Have the authors made all data underlying the findings in their manuscript fully available?

Reviewer #1: Yes

Reviewer #2: Yes

4. Is the manuscript presented in an intelligible fashion and written in standard English?

Reviewer #1: Yes

Reviewer #2: Yes

5. Review Comments to the Author

Reviewer #1: The study titled “Year-round temporal stability of a tropical, urban pollination network” is interesting and a result well-worthy of publication. The presentation is concise and brings out the important conclusions convincingly. However, there are a few problems that should be addressed for clarity and so that the readers fully understand the implications of the results.

One is a fuller description of the parks. Parks can range from relatively “wild” areas that rely on natural precipitation to highly manicured gardens that are regularly watered. As I read the paper, it became increasing evident that the parks in this study are highly manicured (I think this is correct). This should be conveyed in the introduction and stated again in the methods. If the authors could document how many of the plant species they sampled were native or domesticated, that would give the readers an idea of the types of parks sampled in the study. Watering the plants surely decreased the variation in floral abundance and is partly to explain why the pollination networks change so little through the year.

Second, the title states that the study is about temporal changes over the year, but the comparisons of floral abundance, floral species richness, etc. are conclusions that are based on differences across parks. If this is the case, the study is also about spatial changes. This should be addressed.

Finally, and importantly, is choice of plant species sampled at each park. The text is vague on this and needs to provide more detail. Line 80 states that the plants where observations were made were those that were “the most abundant plant species that were in flower”. How was this determined? How many plants were studied at each park and each date? Why did the study use this approach and not a protocol that sampled the same plant species no matter how many flowers each species had? It is important that the protocol used is justified.

This protocol has important implications for what the measures of floral abundance mean. As q=written it appears that your 2 x 2m plots is extrapolated to the size of the entire park which would result in a gross overestimate of actual floral abundance. Please explain all this more carefully and discuss how this choice influences the interpretation of your results, or it does at all.

Below are more detailed comments

L. 22,23- this relates to one of my larger comments above. This study is about tropical pollination networks that are probably more stable than networks in the temperate zone, but also the study is in highly manage parks. That should be made clear at this point.

L. 29. Add that when “all parks are pooled” there is no variation in pollinator abundance or richness.

L. 30-Add that when parks are compared, there is evidence that floral abundance, etc. influence pollinator richness.

L.80. How did you determine when plants were abundant or rare?

L. 86 swing could be deleted

L 93-95. Please be more explicit and complete about how floral abundance and richness are determined, and extrapolated to total park size, etc. How did you minimize bias?

L109. As stated above, you examined how pollinator communities varied over time (month to month comparisons when all parks are polled), and space (given that the number of flowering plants and floral abundance were compared across parks.

L. 112. The use of the term “trophic level” is confusing given that there is only plants and their pollinators. Should this be compartments, or some other term?

L. 198-199-vaguely written. Rewrite for clarity.

L 225. Change “within” to “of”

L 250. either delete or rewrite the phrase “…that are undeterred by human activity.”

L270. This reference is improperly formatted

Add scale bar to map of parks

Reviewer #2: The presented results give an important and new insight into pollinator communities of tropical cities and I think they should be published. However, data collection had some limitations, limiting also the interpretation of data which should be outlined a little more clearly. First of all, when recording insect visitation to plants, you do not have information on whether the insects were pollinating the plants (even when the insect was observed to touch the reproductive parts of the flower). Therefore, I would caution against using the term pollination network. Instead, I would suggest to rather use the term plant insect/pollinator visitation network.

Furthermore, only the most abundant plants in the parks were observed which means that only a fraction of the entire network was selected which also imposes a strong bias for the assessment of network properties. The use of this method also means that the habitat variables floral abundance and floral richness are actually not representative for the park which could easily be misunderstood by the reader. Floral abundance was based on 2x2 m plots with particularly abundant plants. And floral richness included only those most abundant plants. While these parameters can still be related to observed plant visitors, it should be pointed out more clearly for the interpretation of the results.

Additionally, the analyses with linear mixed modelling should be revised and explained more detailed. Your description is a little unclear, but it sounds like you calculated several independent models for each climatic and habitat variable which, in my opinion, is inadequate. Including month as a random factor in some models, although it was not significant as a fixed factor in another model for the same response variable illustrates part of the problem. All climatic and habitat variables that could potentially affect pollinator abundance or richness should be examined in one model (per response variable). You should also consider using the same explanatory variables (as used for pollinator abundance and richness) for the analyses of the network properties. Using the linear mixed model method (LMM) requires normal distribution of the data. Therefore, you should state whether data was normally distributed. The results of the LMM should also include R2-values. The graphs of model predictions (Fig. 2) show always three curves, for low, medium and high predictions (Fig. 2, A-C), except for one of the graphs (Fig. 2, D). It is not explained what these predictions are based on and why Fig. 2, D has only one curve. Typically, there would be just one curve per response variable and explanatory variable. Please either provide additional information or change the graphs.

Another critical point is the evaluated effect of precipitation. The significance of precipitation as a predictor for any response variable is questionable, because all parks were regularly watered additionally to natural precipitation which the model does not take into account. This influence should be considered in analyses and interpretation.

Finally, you state in your conclusion that the studied urban tropical environments host abundant and diverse bee communities. However, you do not relate pollinator abundance or species richness to pollinator communities of natural habitats within the same region. If possible, this information should be added. Otherwise, the value of these urban habitats remains unclear.

Overall, the manuscript is very nicely written and the clear language makes it easy to follow. I hope that you find my comments helpful for revising your manuscript. I think it will be an interesting and valuable addition to the literature.

Minor comments

Abstract

L32 You mention a change of species composition here in the abstract, but do not include this information in the result section. Please add this in the results.

Introduction

L40 Consider adding another more general reference additional to 1 & 2.

Methods

L71 Maybe number of inhabitants and area of Bangkok could be added.

L73-75 What was the size of the parks?

L79-81 What was the range of number of plant species observed per park or in other words how many different plots did you observe per park? Were plots fixed or newly assigned for each observation?

L87 Within how many days did you observe all nine parks each month? Was the order always the same?

L90 “average” instead of “averagely”

L90-93 As you did not measure temperature yourself, I would rephrase: We also obtained data on… from …

L98-116 Please revise your approach of LMM and add details as suggested above.

Results

L130-133 Please explain what points and bars represent.

L135-142 Please indicate whether the listed significant explanatory variables increase or decrease the response variables.

L145-149 Consider making significant predictors bold for easier reading in black and white.

L151-157 Please explain what low, medium and high predictions are based on or change as suggested above.

L154-156 Graph B does not show precipitation as suggested by the caption. Likewise, graph C does not show floral richness nor the interaction of floral abundance and richness.

Discussion

L195 …most orders of insects…

Conclusion

L245-246 Without a comparison to pollinator communities of natural or other kind of habitats in the same region, it is not really possible to make a statement on whether the recorded pollinator community is very abundant or species rich or how valuable of a habitat the parks are.

6. PLOS authors have the option to publish the peer review history of their article (what does this mean?). If published, this will include your full peer review and any attached files.

Reviewer #1: No

Reviewer #2: Yes: Dr. Nicola Seitz

---

## [Author Response · Author response to Decision Letter 0]

11 Dec 2019

Please see the "Response to Reviewers" file.

---

## [Decision Letter · Decision Letter 1]

24 Jan 2020

PONE-D-19-19692R1

Year-round temporal stability of a tropical, urban pollination network

PLOS ONE

Dear Dr. Stewart,

Thank you for submitting your manuscript to PLOS ONE. After careful consideration, we feel that it has merit but does not fully meet PLOS ONE’s publication criteria as it currently stands. Therefore, we invite you to submit a revised version of the manuscript that addresses the points raised during the review process.

This manuscript has been now evaluated by the two previous reviewers. Both of them agree that the revisions of the manuscript have helped to improve considerably the clarity and impact of the paper and that the authors have done a great job with the revision. However, one of the reviewers still have some minor comments and suggestions that I would like the authors to consider. If you disagree with any of the points raised, please explain it in the response letter.

We would appreciate receiving your revised manuscript by Mar 09 2020 11:59PM. To enhance the reproducibility of your results, we recommend that if applicable you deposit your laboratory protocols in protocols.io, where a protocol can be assigned its own identifier (DOI) such that it can be cited independently in the future. For instructions see: http://journals.plos.org/plosone/s/submission-guidelines#loc-laboratory-protocols

We look forward to receiving your revised manuscript.

Kind regards,

Amparo Lázaro, PhD

Academic Editor

PLOS ONE

Reviewers' comments:

Reviewer's Responses to Questions

**Comments to the Author**

1. If the authors have adequately addressed your comments raised in a previous round of review and you feel that this manuscript is now acceptable for publication, you may indicate that here to bypass the “Comments to the Author” section, enter your conflict of interest statement in the “Confidential to Editor” section, and submit your "Accept" recommendation.

Reviewer #1: All comments have been addressed

Reviewer #2: (No Response)

2. Is the manuscript technically sound, and do the data support the conclusions?

Reviewer #1: Yes

Reviewer #2: Partly

3. Has the statistical analysis been performed appropriately and rigorously? 

Reviewer #1: Yes

Reviewer #2: Yes

4. Have the authors made all data underlying the findings in their manuscript fully available?

Reviewer #1: Yes

Reviewer #2: (No Response)

5. Is the manuscript presented in an intelligible fashion and written in standard English?

Reviewer #1: Yes

Reviewer #2: Yes

6. Review Comments to the Author

Reviewer #1: The manuscript is much clearer and more concise. I have no comments and recommend it for publication.

Reviewer #2: The revised manuscript is much improved and most concerns were addressed and changes performed accordingly. A few unclarities and concerns remain and I would suggest some more minor revisions.

Following up on 1. of my previous comments and the authors reply: Although some studies use the term pollination network, the use of this terminology is problematic and actually incorrect as research has shown that pollination cannot be inferred from flower visitation (see e.g. King et al. 2013 or Popic et al. 2013). Less problematic is the use of plant pollinator networks which I suggest as one alternative to pollination networks and which numerous other studies use (including many you have cited, i.e. Robinson et al. 2018, Alarcón et al. 2008 or CaraDonna et al. 2017). However, the most accurate term is plant insect networks (used e.g. by Vanbergen et al. 2014 or Losapio et al. 2015). For improvements in the fields of pollination ecology and plant insect interactions, I would encourage the use of plant insect networks when visitation and not pollination was assessed like in this study.

Following up on 2. of my previous comments and the authors reply: The sampling methods for determining floral abundance and richness were clarified. However, the resulting bias and limitations for the interpretation of results is still not sufficiently discussed in my opinion. The described selective sampling only from the most abundant flowers creates a bias for network analyses, because plants present in low numbers are consistently missing. Please consider this bias in the discussion. Furthermore, the variables floral abundance and floral richness are not representative for the actual park vegetation as they are based on the most abundant patches of flowers which is a selective measure. Please make this more clear in the discussion to avoid confusion.

L89 It still remains unclear if only one park a day was observed, all in one day or how many parks per day within what time span of a month (e.g. three parks a day, and all parks within first week of the month).

L96 visitation interactions instead of pollination interaction

L112-113 If only those flowers were observed that had a minimum of 20 flowers/inflorescences this calculation does not really reflect the floral abundance of the park.

L164-167 Suggestion for consistency/easy reading: either individual brackets with “(positive effect)” always after the first and second predictor or “(positive effects)” only after the second predictor.

L180-189/Fig. 2: I find the interpretation of the graphs still a little difficult. Can you provide more information on the visualization of the model predictions in the methods section?

L272-278 As mentioned above, you should point out that the visitation networks analyzed completely omitted less abundant plants which is potentially affecting network properties.

L289-290 Maybe this could be rephrased slightly, because considering the much lower species diversity compared to the below mentioned fruit orchards, the pollinator community in the parks does not appear to have a very high level of diversity.

References:

King, C., Ballantyne, G. and Willmer, P.G. (2013), Why flower visitation is a poor proxy for pollination: measuring single‐visit pollen deposition, with implications for pollination networks and conservation. Methods Ecol Evol, 4: 811-818. doi:10.1111/2041-210X.12074

Popic, T.J., Wardle, G.M. and Davila, Y.C. (2013), Flower‐visitor networks only partially predict the function of pollen transport by bees. Austral Ecology, 38: 76-86. doi:10.1111/j.1442-9993.2012.02377.x

Robinson, Samuel VJ, Gianalberto Losapio, and Gregory HR Henry. "Flower-power: Flower diversity is a stronger predictor of network structure than insect diversity in an Arctic plant–pollinator network." Ecological complexity 36 (2018): 1-6.

Alarcón, Ruben, Nickolas M. Waser, and Jeff Ollerton. "Year‐to‐year variation in the topology of a plant–pollinator interaction network." Oikos 117.12 (2008): 1796-1807.

CaraDonna, P. J., Petry, W. K., Brennan, R. M., Cunningham, J. L., Bronstein, J. L., Waser, N. M., & Sanders, N. J. (2017). Interaction rewiring and the rapid turnover of plant–pollinator networks. Ecology letters, 20(3), 385-394.

Vanbergen, A.J., Woodcock, B.A., Gray, A., Grant, F., Telford, A., Lambdon, P., Chapman, D.S., Pywell, R.F., Heard, M.S. and Cavers, S. (2014), Grazing alters insect visitation networks and plant mating systems. Funct Ecol, 28: 178-189. doi:10.1111/1365-2435.12191

Losapio, G., Jordán, F., Caccianiga, M., & Gobbi, M. (2015). Structure-dynamic relationship of plant–insect networks along a primary succession gradient on a glacier foreland. Ecological modelling, 314, 73-79.

7. PLOS authors have the option to publish the peer review history of their article (what does this mean?). If published, this will include your full peer review and any attached files.

Reviewer #1: No

Reviewer #2: Yes: Dr. Nicola Seitz

---

## [Author Response · Author response to Decision Letter 1]

28 Jan 2020

Please see the uploaded "Response to Reviewers" file.

---

## [Editor Report · Decision Letter 2]

3 Mar 2020

Year-round temporal stability of a tropical, urban plant-pollinator network

PONE-D-19-19692R2

Dear Dr. Stewart,

We are pleased to inform you that your manuscript has been judged scientifically suitable for publication and will be formally accepted for publication once it complies with all outstanding technical requirements.

With kind regards,

Amparo Lázaro, PhD

Academic Editor

PLOS ONE
---

## [Editor Report · Acceptance letter]

25 Mar 2020

PONE-D-19-19692R2 

Year-round temporal stability of a tropical, urban plant-pollinator network 

Dear Dr. Stewart:

I am pleased to inform you that your manuscript has been deemed suitable for publication in PLOS ONE. Congratulations! Your manuscript is now with our production department. 

With kind regards,

on behalf of

Dr. Amparo Lázaro 

Academic Editor

PLOS ONE